# Factors Affecting Psychological and Health-Related Quality-of-Life Status in Children and Adolescents with Congenital Heart Diseases

**DOI:** 10.3390/children9040578

**Published:** 2022-04-18

**Authors:** Hao-Chuan Liu, Chung-Hsien Chaou, Chiao-Wei Lo, Hung-Tao Chung, Mao-Sheng Hwang

**Affiliations:** 1Division of Cardiology, Department of Pediatrics, Chang Gung Memorial Hospital, Linkou Branch, Taoyuan City 333, Taiwan; hungtao@cgmh.org.tw (H.-T.C.); hms3013@cgmh.org.tw (M.-S.H.); 2Department of Emergency Medicine, Chang Gung Memorial Hospital, Linkou Branch and Chang Gung University College of Medicine, Taoyuan City 333, Taiwan; shien@url.com.tw; 3Department of Pediatrics, Cathay General Hospital, Taipei Branch, Taipei City 106, Taiwan; bluesnowfalling@gmail.com

**Keywords:** congenital heart disease, children and adolescents, depression, health-related quality of life, questionnaire

## Abstract

Congenital heart disease (CHD), a severe cardiac defect in children, has unclear influences on young patients. We aimed to find the impacts of differently structure heart defects and various treatments on psychology and health-related quality of life (HRQoL) in CHD children and adolescents. CHD patients aged between 6 and 18 years old visited our hospital from 1 May 2018 to 31 September 2018, and their principal caregivers were asked to participate. We used two validated questionnaires, Children Depression Inventory-TW (CDI-TW) and Child Health Questionnaire—Parent Form 50 (CHQ-PF 50), to evaluate CHD patients’ psychological and HRQoL conditions. Participants were grouped based on their cardiac defects and previous treatments. We analyzed the results via summary independent-samples t-test with post hoc Bonferroni correction and multivariant analysis. Two hundred and seventy-seven children and their principal caregivers were involved. There was no apparent depressive condition in any group. Single cardiac defect patients exhibited similar HRQoL to controls; simultaneously, those with cyanotic heart disease (CyHD), most multiple/complex CHDs children and adolescents, and those who received invasive treatments had poorer HRQoL. CyHD impacted the most on patients’ psychological and HRQoL status. Patients with sole cardiac defect could live near-normal lifes; on the other hand, CyHD had the worst effects on patients’ psychology and HRQoL.

## 1. Introduction

Congenital heart disease (CHD) is the most common type of congenital disability. More than one million CHD children have been born worldwide. With advances in the treatment of congenital and acquired heart disease, the survival rate of CHD children has improved substantially over the years. More than 90% of CHD children could reach adulthood [1,2]. Due to the improving prenatal diagnosis and advanced treatments, the number of adults with CHDs (ACHDs) will eventually exceed the number of children with CHD in the future. Recent studies have shown elevated risks for physical disability, psychological distress, and social challenges in ACHD survivors [3,4]. Before CHD children reach adulthood, understanding their psychological condition is essential for medical and mental health professionals to provide adequate support and achieve more holistic care.

Many ACHDs have experienced significant emotional distress, including anxiety symptoms and depression [5]. Depression is increased significantly in ACHDs compared with controls [6]. This might be due to their cardiac defects which need long-term medical care, repeat hospitalization, and the uncertainty of future interventions. The relationship between depression and poor cardiovascular (CV) outcomes has been proposed for acquired heart disease patients [7]. Jackson et al. stated that chronic emotional distress might increase CV complications and premature mortality in ACHDs; however, emotional distress’s prevalence and effect are still unknown in CHD children and adolescents [8].

Quality of life (QoL) is “an overall general well-being that comprises objective descriptors and subjective evaluations of physical, material, social, and emotional well-being together with the extent of personal developmental and purposeful activity, all weighted by a personal set of values“ [9]. It includes physical and psychosocial components. Physical status indicates one’s ability to perform various physical activities, self-care behaviors, and other tangible standards, whereas psychosocial status suggests one’s function in school, social, and emotional domains [10]. Health-related quality of life (HRQoL) is one’s QoL which is “affected by the presence of disease or treatment” [11]. Factors that cause poor QoL in CHD patients include higher frequency and severity of clinical symptoms, the severity of CHDs, the need for interventions, poor functional status, repeat hospitalizations, impaired general health status, and the number of inter-familial conflicts [12,13,14,15].

Premature morbidity and mortality are higher in many CHD survivors due to growing anxiety and depression with time. The depressed mood had been shown to correlate with poor QoL in ACHDs positively [5,13]. Identifying factors that cause depression and impaired QoL in younger CHD patients might improve their outcomes [16]. Previous studies usually divided patients into moderate and complex CHDs or whether they had cyanosis instead of more detailed CHD classifications and have had inconsistent results [5,12,13,14,15,17]. Given the heterogeneity of cardiac lesions in CHD, the variability in disease burden and lesion severity could be critical factors to different psychological and QoL outcomes. Our study uses a prospective method to evaluate disparate CHDs according to their hemodynamic characteristics and treatment methods. By using more precise classifications, we can obtain a more detailed cause of depression and HRQoL in young age CHD patients.

## 2. Materials and Methods

Three hundred and fifty patients with structural CHDs visited our hospital as in-patients or out-patients from 1 May to 31 September 2018, and one major caregiver each was asked about participation. Inclusion criteria were (1) confirmed diagnosis by either echocardiography, cardiac catheterization, or open-heart surgery; (2) aged between 6 and 18 years old; (3) no diagnosis with cardiomyopathies or simple congenital valvular regurgitations; (4) accompaniment by one major caregiver, such as a parent, who can read and write; and (5) both the patient and the caregiver being willing to join the study. Exclusion criteria: either the patient or the caregiver were (1) intellectually disabled, (2) unable to read or understand the questionnaire, or (3) unable to finish the questionnaire. Questionnaire of Children Depression Inventory-TW (CDI-TW) was given to the CHD patient or gone through with a research assistant if needed. Child Health Questionnaire—Parent Form 50 (CHQ-PF 50) was given to the caregiver simultaneously. We separated patients and their caregivers while filling out the questionnaire, if possible, to avoid untruthful answers. We reviewed their clinical data, including current age, gender, diagnosis, if the patient had been admitted to the intensive care unit (ICU), date and age of receiving operation, date of previous cardiac catheterization, and date of prior therapeutic cardiac catheterization. We categorized patients according to their management, i.e., admitted to ICU (ICU), received cardiac catheterization (CC), received therapeutic catheterization (TCC), and had operations (OP). The complexity of their CHD condition classified patients into simple CHDs (sCHD), which include pure atrial septal defect (pASD), pure ventricular septal defect (pVSD), pure right-side great arteries anomalies (pRGA), pure left-side great arteries anomalies (pLGA), and pure patent ductus arteriosus (pPDA); and multiple/complex CHD (mCHD), which include multiple/complex CHDs with atrial septal defect (mASD), with ventricular septal defect (mVSD), with right-side great arteries anomalies (mRGA), with left-side great arteries anomalies (mLGA), with venous return anomalies (mVR), or with patent ductus arteriosus (mPDA); and cyanotic heart diseases (CyHD).

### 2.1. Children Depression Inventory-TW

Children Depression Inventory-TW (CDI-TW) is a Chinese version of the Children Depression Inventory, the most widely validated tool used to evaluate children’s depression [18]. The psychometrics in CDI-TW have been examined and validated on Taiwanese youth [19]. CDI-TW is a self-reported questionnaire with 27 different questions to assess various depressive symptoms. Each question has three options indicating different severity of depressive symptoms: 0, 1, and 2. The results are calculated and presented as other depressive aspects, including negative mood (NM), interpersonal problems (IP), ineffectiveness (IN), anhedonia (AN), and negative self-esteem (NE). Adding different aspects would generate a CDI-TW total score (CDI-TW score). A higher score means a more severe depressive condition. CDI total score of more than 20 is used as a cut-off score for detecting depressed patients [20]. We compare the results scores in each element and the total score with those of normal children with same-aged group in Taiwan. We also performed multivariant analysis for CDI-TW score to discover the correlation between CDI-TW and Child Health Questionnaire-parent Form 50 (CHQ-50PF).

### 2.2. Child Health Questionnaire—Parent Form 50

Child Health Questionnaire—Parent Form 50, a validated parent-proxy questionnaire, scores for 14 concepts (12 multi-item scales and two single items) using the major caregiver of 5 to 18 year-old patients. According to their latest four weeks’ experience, they answered the questionnaire to explain patients’ health-related quality of life (HRQoL) [21]. The evaluating concepts include physical functioning (PF), role/social-physical (RP), general health perceptions (GH), bodily pain (BP), role/social-emotional/behavioral (REB), parental impact-time (PT), parental impact-emotional (PE), self-esteem (SE), mental health (MH), behavior (BE), family activities (FA), family cohesion (FC), and change in health (CH). Previous reports have validated PF, RP, GH, and BP as a measure of physical health status; and REB, PE, SE, MH, BE, and FC scales are the best measurements for psychosocial conditions. A lower score indicates a more limited physical or psychosocial health condition. Physical summary score (PhS) and psychosocial summary score (PsS) are scored according to 11 concepts, but the scoring manual excluded FA and FC because they are still being worked on. At the same time, we calculated the Phs and PsS total scores according to the scoring manual, compared with scale norms from United States’ general young population, and used PhS and PsS scores for further multivariant analysis.

### 2.3. Statistics

To understand the differences between our patients and the normal population, we compared different aspects of CDI-TW and CHQ-50PF with the aspects of healthy references. We analyzed continuous variables with a summary independent-samples t-test and performed Bonferroni correction as post hoc analysis. *p* ≤ 0.001 was defined as statistically significant for these analyses. Furthermore, we analyzed the relationships between CDI-TW score, PhS, and PsS under different conditions to understand their effects on depression and HRQoL in CHD children and adolescents by performing multivariant analysis. The multivariant analysis also analyzed the relationships among PsS, PhS, and CDI-TW to understand the relationship between depressive mood and HRQoL in CHD children and adolescents. The statistical analysis was performed via IBM SPSS Statistics 25 (IBM Corp, New York, NY, USA) and SAS 9.4 (SAS Institute Inc., Cary, NC, USA). *p* ≤ 0.05 was defined as statistically significant in the multivariant analysis.

## 3. Results

A total of 277 patients aged between 9–18 years old (or their major caregivers) completed and returned the questionnaires. The patients’ mean age was 15.05 years old. Demographic results are shown in Table 1.

In the CDI-TW questionnaire, while comparing to the normative sample, all groups with pure, multiple/complex cardiac defects and various treatments showed no sign of significant depression in any aspect. However, CyHD children and adolescents had significantly negative attitudes toward their abilities and school performances (statistically significantly higher IN) than norms (Table 2).

In the CHQ-50PF questionnaire, CyHD children and adolescents showed significantly lower scores in almost all the aspects, including FA, PT, nearly all physical metrics (PF, RP, GH scales), and half of the psychosocial (REB, PE, and SE scales) health components. In other words, their physical and psychosocial status had deteriorated significantly, including most of the contributing factors. However, their mental health, general behavior, bodily pain, and family relationships were normal (significantly lower PhS and PsS but normal BP, MH, BE, and FC scales). All the sCHD children and adolescents were generally considered in poor health (lower GH); however, sASD and sVSD had less bodily pain than controls (higher BP) (Table 3). Major caregivers also considered that all the mCHD children and adolescents had poorer health (significantly lower GH). The questionnaire results showed that mVSD and mRGA children had similar scale distributions, with worse self-esteem, physical performance, family activity, parent time, and emotions. The decreased family activity, parent time, and emotions caused limitations in their daily lives and family relationships (statistically significantly lower SE, PF, FA, PT, and PE). In other mCHDs, mPDA had the third most abnormal scales compared to other mCHDs. In addition to health concerns (decreased GH), mASD children and adolescents had more behavioral problems (lower BE); their parents also felt much emotional stress about their health (significantly lower PE). However, they had less bodily pain than the standard samples (significantly lower BP). Groups of mVR had the least abnormal scales compared to the other mCHDs. Besides general health concerns, parents of mLGA and mPDA experienced more time limitations and emotional stress (significantly lower PT and PE). mPDA children and adolescents had less self-esteem than norms (significantly lower SE). Meanwhile, children and adolescents with mVR had fewer emotional problems in school (a considerably higher REB) than norms. Except for mVR children and adolescents, all the mCHD children and adolescents were impaired psychosocially (lower PsS score) compared to the normative sample. At the same time, mVSD and mRGA children and adolescents also had significantly worse physical condition (lower PhS score) related to HRQoL (Table 3).

For CHD children and adolescents who received treatments, ICU and OP groups were in worse psychosocial condition, and had physical limitations, poor general health, decreased self-esteem, immature behavior, interrupted family activity, limited parental time, and emotional stress (significantly lower PsS with lower PF, GH, SE, BE, FA, PT, and PE) compared to the normative sample; meanwhile, the CC group had similar results except a regular physical activity (normal PF). Furthermore, OP also were in significantly poorer psychosocial condition (lower PhS). The TCC group was considered to have poor health. Their parents experienced time limitations and emotional stress due to their health conditions (significantly lower GH, PT, and PE). All the treatment groups exhibited less bodily pain (significantly higher BPs) than norms (Table 3).

Having CyHD and VSD significantly affected patients’ depressive condition (CDI-TW) among all CHD patients in multivariant analysis (Table 4). The presence of CyHD caused a higher CDI-TW score compared to non-CyHD (beta: 5.62, SE: 2.0, *p* = 0.004); instead, having a VSD resulted in a lower CDI-TW score than not having a VSD (beta: −4.059, SE: 1.934, *p* = 0.037). The patients’ physical condition (PhS score) was significantly related to CyHD. Having stayed in the ICU, and CHD children and adolescents only having ASD caused lower PhS total scores. Moreover, the factors of CyHD and having stayed in ICU also significantly caused significantly lower PsS scores. When evaluating the relationship between patients’ depression, physical condition, and psychosocial condition, the depressive mood was significantly related to poorer psychosocial condition among CHD patients. (beta: −0.33, SE: 0.06, *p* < 0.0001) (Table 5).

## 4. Discussion

Our study presented the effects of detailed cardiac defects and management on psychological status and HRQoL in CHD children and adolescents. We also showed the relationships between depression, physical, and psychosocial conditions within those patients. Cyanotic heart disease children demonstrated “ineffectiveness,” a depressive scale, and the worst HRQoL with impaired physical and psychosocial healthiness. Their health condition caused physical limitations and emotional changes, resulting in poor daily functioning and significantly impacting their personal lives, family lives, and school lives. Moreover, children and adolescents with CyHD had poorer self-esteem, their parents worried about their health more, and the disease impacted their parents’ lives. Multiple/complex CHDs, such as mVSD and mRGA, and those who received interventions, also showed similar results. Among received treatment groups, those who underwent therapeutic catheterization had the best HRQoL without obvious physical and psychosocial impairments. Only their parents’ lives were affected. On the contrary, those who received operations had significantly impaired physical and psychosocial status, along with CC and ICU stay groups. Their health condition afflicted their school and family lives, causing poorer self-esteem and more aggressive behavior. None of the groups showed depressive or mental health impairments in our study. Bodily pain was not seen in CHD children and adolescents and was even better or equal to the norms in most groups. Our results provide a more detailed overview and reflect the previous finding that more complex CHD, more hospital admissions, and more cardiac surgeries lead to poor QoL in CHD children and adolescents [22].

There was no difference between self-reporting and parent-proxy results regarding anxiety and depression in this study. For CDI-TW, a self-report questionnaire, there was no statistically significant difference in CDI-TW total score, a generally depressive symptom score, among CHD groups. Additionally, there was no statistically significant decreased MH, which represents a feeling of anxiety and depression, according to CHD-50PF, a parent-proxy report questionnaire. Varni et al. proved the reliability and validity of parent-proxy reporting in children [23,24]. The self and parent-proxy responses to the questionnaires are correlated [22]. Multiple reports have shown that the severity of CHD has positive relationships with depressive and psychological conditions [25,26]. Since a higher score in CDI-TW indicates a more severe depressive condition, almost all the scores in CDI-TW were high, including incredibly significantly higher IN, in CyHD, which is the most severe CHD.

Children’s depressiveness and HRQoL is related to disease severity and condition after treatment. Simple CHDs had no depressive indicators and fewer QoL concerns. There were noticeable scale differences between sCHD and mCHD, except for mVR and those who received treatments. There were fewer significantly different scales using the CHQ-50PF in mVR, whose QoL scale scores were nearly the same as those of the sCHD groups. In our study, the most common disease in mVR was partial anomalous pulmonary venous return anomalies (PAPVR), classified as multiple cardiac defects, because it always has at least one more cardiac defect, i.e., an atrial defect. PAPVR children usually do not have heart symptoms or need to have them corrected at a very young age [8]. It was reasonable for mVR patients to have similar results to sCHDs. At the questionnaire age, they usually have no or only minimal heart symptoms. Children with venous return anomalies did not have a depressive condition. They had average general HRQoL, or even improved activity regarding their motions or behavior; however, their parents still believed their children’s health was poor.

In children with more severe CHDs, such as CyHD, those who had multiple cardiac defects, and those who received treatments, their parents reported limited personal time, limited family activity, and experienced a great deal of emotional stress due to their children’s condition. When taking care of CHD patients, parents bore more psychological stress, depression, and anxiety because of the uncertain future [27,28]. Parental mental health is crucial regarding HRQoL in CHD children [29]. More severe CHDs would cause worse parental health than sCHDs and worsen their HRQoL. In contrast, mVR, also known as mCHDs, had fewer impacts on parental mental health. When taking care of those mCHDs, parents had less or the same stress as those taking care of healthy children. Therefore, the parents have mental health status just like other ordinary parents.

Our results showed generally significantly decreased general heath in every group. There was no bodily pain or discomfort in CHD children and adolescents; surprisingly, CHD children felt less bodily pain/discomfort than average children. Instead of musculoskeletal pain, chest pain due to arrhythmia and heart failure were the most frequent painful sensations related to the heart in adult congenital heart disease patients. Generally, people regard CHDs as severe diseases in children due to their high mortality and morbidity. Parents usually suffer from much stress about their children’s health. Even if the cardiac defect is not hemodynamically significant or has no severe symptoms before or after the surgery, parents still worry more about children’s health. Due to parental concern, CHD children usually receive better care and even overprotection unrelated to the condition’s severity [30,31]. In addition, their physical activity was usually limited [32,33]. Those factors would further decrease the chance of developing musculoskeletal pain or physical discomfort.

There was lower self-esteem in most CHD children and adolescents except in sCHD patients, relatively simple mCHD children, and TCC patients. Self-esteem is a personal characteristic that facilitates a positive perception of stress [34]. Poor self-esteem has been related to poor QoL and depression [35,36]. Lower self-esteem was previously found in CHD patients and correlated with disease severity, but improved after an operation [37,38]. Our results partially reflect previous results, though the poor self-esteem did not recover even received treatments, especially the open-heart surgery. That might have been caused by the relatively unstable condition of CHD children and the need for long-term following up. sCHDs, mASD, mLGA, mVR, and TCC children and adolescents presented good self-esteem in line with the normal population.

Cyanotic heart disease is the most severe structural cardiac defect that presents from birth. Disease severity and the perceived disease severity have been reported to be correlated with patients’ depressiveness, psychosocial condition, and HRQoL [16,26,37,39,40,41,42]. CyHD children were aware of being different from other children due to regular and frequent hospital visits. The divergence between them and other children would affect their relationships and cause impaired psychosocial bonds. Strong friendship bonding could also be harder for them owing to their cyanotic appearance. These results affect their emotional, academic, cognitive, and behavioral functioning and lead to a more depressive mood and worse QoL. Besides disease severity, HRQoL is also impacted by the cardiac symptoms [15], and most evident cardiac symptoms need ICU treatment. In our multivariant analysis, having CyHD, the most severe CHD with the most prominent cardiac symptoms, was the main factor for a worse CDI-TW score, poor physical condition, and poor psychosocial status according to CHQ-50PF. Having previous ICU stays was also a factor for poor physical and psychosocial scores.

In our multivariant analysis, physical health was the factor that most affected CDI-TW total score, which indicates an individual’s depressive status. Increased physical activity could reduce or prevent depressive symptoms and major depressive disorder [43,44]. Morgan et al. described that ceasing all forms of vigorous physical activity can cause an individual to develop depressive symptoms [45]. Exercise capacity was also shown to predict future mortality and morbidity in CHD patients [46]. With the importance of physical exercise, guidelines have included effective exercising in treatment guidelines for affective disorders [47]. Our results provide evidence that physical activity is essential and directly relates to the depressiveness in CHD children.

The strengths of this study included a sizeable total sample size, a high parent-patient paired response rate, an accurate disease classification, and using validated questionnaires with normative data across a wide age span. The study’s limitations were a lack of broad geographic, racial, and ethnic diversity, a cross-sectional design, and a single medical center. In addition, CDI-TW could only pick up the most severely depressed patients and has poor sensitivity to subtle and non-clinical conditions. Moreover, CHQ-50PF is a generic instrument instead of a cardiac-specific tool like the Pediatric Cardiac Quality of Life Inventory (PCQLI) [48]. However, PCQLI does not have a validated Chinese translation. Furthermore, CHQ-50PF is a parent-proxy questionnaire that could cause a bias due to a potential difference between children and parent proxy reports [49]. Further investigations could focus on the impacts of QoL and depressive conditions on different CyHDs by using a cardiac-specific tool and insert more broad measurements for psychological functioning. Future analysis could also use a more detailed classification of age and gender to see the differences. Moreover, with the development of molecular biology, there are several new methods with which to understand people’s genetics, such as whole-genome sequencing. With the new methods, one could also find the relationships between gene variants and different cardiac defects and related depressive and HRQoL conditions.

## 5. Conclusions

Our study showed impaired psychological condition and decreased HRQoL were more prevalent in children and adolescents who had CyHD, had received treatments, or had mCHD, except for those who were mVR. Among them, children and adolescents who received treatments, except for operations, had very few different physical status indicators from the normative sample. CyHD had the most impact on children’s depressive mood and HRQoL. Simple CHD and mVR children could have a near-normal depressive and HRQoL status. Due to the advances in CHD treatments, there are rising numbers of CHD children reaching adolescence and adulthood, which is causing an increasing global burden [50]. Identifying the impaired HRQoL in CHD children might potentially improve the outcomes [16]. For better patient care, we should pay more attention to psychological and HRQoL condition in children with cyanotic heart diseases, in those with more complex congenital heart diseases, and in those who require management, since they are young, and encourage their families to let them live normal lives. For sCHD and mVR children and adolescents, we do not need to over-worry about their depressive and QoL conditions. We should treat them as near-normal children.

## Figures and Tables

**Table 1 children-09-00578-t001:** Demographic data of children and adolescents with congenital heart disease.

	Simple Congenital Herat Diseases						Multiple/Complex Congenital Heart Disease					
	pASD	pVSD	pRGA	pLGA	pPDA	CyHD	mASD	mVSD	mRGA	mLGA	mVR	mPDA
Number	66	58	19	12	11	48	65	82	53	11	9	20
Mean age (year-old) (SD)	13.3 (3.2)	13.8 (3.3)	15.1 (2.7)	13.5 (3.4)	12 (3.8)	13.7 (3.0)	13.1 (3.5)	15.6 (2.8)	13.6 (3.1)	13.6 (3.9)	14.7 (3.2)	13.4 (4.2)
Male/female	25/41	31/27	9/10	9/3	5/6	24/24	28/37	42/40	29/24	6/5	4/5	11/9
Numbers had stayed in ICU (%)	38 (58%)	30 (52%)	2 (11%)	5 (42%)	2 (18%)	37 (77%)	50 (77%)	73(89%)	41 (77%)	9 (82%)	8 (89%)	19(95%)
Numbers had received catheterization (%)	46 (70%)	38 (66%)	13 (68%)	7 (58%)	9 (82%)	38 (79%)	55(85%)	73(89%)	49 (92%)	8 (72%)	8 (89%)	18 (90%)
Number had received therapeutic catheterization (%)	34 (52%)	4 (7%)	11 (58%)	6 (50%)	9 (82%)	5 (10%)	15(23%)	9 (11%	7 (13%)	2 (18%)	1 (11%)	3 (15%)
Numbers had received operation(%)	13 (20%)	29 (50%)	0 (0%)	5 (42%)	3 (27%)	43 (90%)	50 (77%)	79 (96%)	46 (87%)	11 (100%)	8 (89%)	20(100%)
Average age when operated (year-old) (range)	3.8 (0–14)	1.4 (0–9)	Nil	0	0	1.4 (0–13)	0.9 (0–6)	0.6 (0–12)	1.0 (0–12)	1.27 (0–14)	0.4 (0–2)	0.8 (0–14)
Average operation times within those had operation (range)	1 (1)	1.1 (1–3)	0	1.4 (1–2)	1 (1–1)	1.7 (1–4)	0.9 (1–2)	1.40 (1–4)	1.5 (1–4)	1.5 (1–2)	1.3 (1–3)	1.6 (1–4)

Abbreviations: Pure atrial septal defect (pASD), pure ventricular septal defect (pVSD), pure right-side great arteries anomalies (pRGA), pure left-side great arteries anomalies (pLGA), pure patent ductus arteriosus (pPDA), multiple/complex CHDs with atrial septal defect (mASD), multiple/complex CHDs with ventricular septal defect (mVSD), multiple/complex CHDs with right-side great arteries anomalies (mRGA), multiple/complex CHDs with left-side great arteries anomalies (mLGA), multiple/complex CHDs with venous return anomalies (mVR), multiple/complex CHDs with patent ductus arteriosus (mPDA), cyanotic heart diseases (CyHD).

**Table 2 children-09-00578-t002:** Children Depression Inventory-TW scores in children and adolescents with congenital heart disease.

		Simple Congenital Herat Diseases					Multiple/Complex Congenital Heart Disease							Received Treatments			
	Norm (n = 1148)	pASD (N = 66)	pVSD (N = 58)	pRGA (N = 19)	pLGA (N = 12)	pPDA (N = 11)	CyHD (N = 48)	mASD (N = 65)	mVSD (N = 82)	mRGA (N = 53)	mLGA (N = 11)	mVR (n = 9)	mPDA (N = 20)	ICU (N = 167)	CC (n = 207)	TCC (N = 80)	OP (N = 146)
CDI-TW Mean(SD)	9.98 (7.29)	9.24 (5.93)	8.79 (5.7)	7.79 (4.81)	9 (5.66)	10.91 (8.09)	11.81 (6.88)	9.74 (6.59)	9.63 (5.95)	10.17 (6.34)	12.18 (6.55)	7.67 (7.12)	9.95 (6.70)	9.8 (6)	9.64 (6.08)	9.29 (5.93)	10.02 (6.16)
NM Mean(SD)	2.21 (2)	1.76 (1.47)	1.9 (1.62)	1.47 (1.17)	2 (2.59)	2.55 (1.86)	2.19 (2.19)	1.83 (1.70)	1.76 (1.47)	1.92 (1.54)	2.09 (1.64)	1.78 (1.64)	1.75 (1.62)	1.84 (1.48)	1.86 (1.56)	1.79 (1.58)	1.87 (1.54)
IP Mean(SD)	0.77 (1.22)	0.77 (1)	0.66 (0.76)	0.26 (0.56)	0.92 (0.79)	1.55 (2.16)	1.04 (1.17)	1.05 (1.18)	0.82 (0.92)	0.77 (0.97)	1.73 (1.19)	0.89 (1.27)	1.20 (1.32)	0.83 (0.99)	0.85 (1.08)	0.88 (1.29)	0.87 (0.97)
IN Mean(SD)	1.9 (1.82)	2.08 (1.83)	1.59 (1.61)	1.79 (1.47)	1.92 (1.78)	1.64 (1.03)	2.77 (1.59) *	2.09 (1.70)	2.34 (1.69)	2.30 (1.69)	2.82 (1.47)	1.44 (1.24)	2.25 (1.55)	2.33 (1.72)	2.17 (1.73)	2.13 (1.68)	2.32 (1.73)
AN Mean(SD)	3.28 (2.66)	3.09 (2.43)	2.88 (2.54)	2.37 (2.19)	2.33 (2.1)	3 (2.93)	3.85 (2.95)	2.71 (2.49)	2.99 (2.57)	3.42 (2.71)	3.55 (2.42)	2.44 (2.3)	3.35 (2.28)	3.07 (2.49)	2.98 (2.44)	2.83 (2.24)	3.1 (2.57)
NE Mean(SD)	1.82 (1.82)	1.55 (1.25)	1.78 (1.33)	1.89 (1.33)	1.83 (1.59)	2.18 (1.78)	1.98 (1.62)	2.06 (1.55)	1.74 (1.23)	1.77 (1.35)	2.00 (1.61)	2 (1.66)	1.40 (1.23)	1.77 (1.38)	1.81 (1.41)	1.68 (1.35)	1.86 (1.43)
Depressed Patients ^&^ Number(%)	Nil	4 (6.06)	4 (6.90)	1 (5.26)	0 (0)	1 (9.09)	7 (14.58)	5 (7.69)	5 (6.10)	5 (9.43)	1 (9.09)	1 (11.11)	1 (5.00)	11 (6.59)	14 (6.76)	4 (5.00)	11 (7.53)

Abbreviations: CDI-TW total score (CDI-TW), negative mood (NM), interpersonal problems (IP), ineffectiveness (IN), anhedonia (AN), negative self-esteem (NE), pure atrial septal defect (pASD), pure ventricular septal defect (pVSD), pure right-side great arteries anomalies (pRGA), pure left-side great arteries anomalies (pLGA), pure patent ductus arteriosus (pPDA), multiple/complex CHDs with atrial septal defect (mASD), multiple/complex CHDs with ventricular septal defect (mVSD), multiple/complex CHDs with right-side great arteries anomalies (mRGA), multiple/complex CHDs with left-side great arteries anomalies (mLGA), multiple/complex CHDs with venous return anomalies (mVR), multiple/complex CHDs with patent ductus arteriosus (mPDA) cyanotic heart diseases (CyHD), patient had admitted to ICU (ICU), patients had received cardiac catheterization (CC), patients had received therapeutic catheterization (TCC), patients had received operations (OP). ^&^ Depressed patients are defined as CDI-TW more than 20. * *p* ≤ 0.001.

**Table 3 children-09-00578-t003:** Child Health Questionnaire—Parent Form 50 scores in children and adolescents with congenital heart disease.

		Simple Congenital Herat Diseases					Multiple/Complex Congenital Heart Disease							Received Treatments			
	Norm (n = 391)	pASD (N = 66)	pVSD (N = 58)	pRGA (N = 19)	pLGA (N = 12)	pPDA (N = 11)	CyHD (N = 48)	mASD (N = 65)	mVSD (N = 82)	mRGA (N = 53)	mLGA (N = 11)	mVR (n = 9)	mPDA (N = 20)	ICU (N = 167)	CC (n = 207)	TCC (N = 80)	OP (N = 146)
PF Mean (SD)	96.1 (13.9)	96.75 (6.64)	94.01 (16.61)	98.82 (3)	86.88 (16.8)	96.45 (11.76)	82.02 (19.9) *	91.79 (14.58)	87.56 (18.11) *	84.88 (19.5) *	93.38 (12.94)	95 (9.05)	90.77 (18.14)	91.5 (16.03) *	92.68 (14.58)	94.26 (15.59)	90.62 (15.23) *
RP Mean (SD)	93.6 (18.6)	97.17 (8.71)	95.08 (17.74)	100 (0)	90.08 (15.31)	93.91 (20.2)	81.7 (22.71) *	90.90 (18.13)	86.83 (21.96)	85.66 (22.23)	84.68 (24.3)	98.11 (5.67)	81.48 (26.07)	90.6 (19.49)	91.93 (18.14)	94.31 (17.08)	89.82 (19.22)
GH Mean (SD)	73 (17.3)	61.74 (14.77) *	61.98 (13.57) *	58.16 (12.72) *	50 (13.98) *	49.09 (20.23) *	50.21 (16.04) *	57.62 (13.7) *	53.54 (15.51) *	51.60 (14.7) *	50.45 (15.72) *	50.56 (9.17) *	50.50 (18.27) *	56.26 (15.28) *	56.98 (15.46) *	57.31 (16.36) *	55.31 (15.24) *
BP Mean (SD)	81.7 (19)	96.06 (10.21) *	90.52 (13.43) *	92.63 (14.08)	93.33 (16.14)	92.73 (9.05)	88.33 (15.89)	91.69 (15.67) *	91.22 (16.05) *	90.75 (15.17) *	86.36 (21.11)	86.67 (13.23)	93.00 (17.5)	92.34 (14.43) *	92.85 (13.51) *	94.38 (12.31) *	92.05 (14.52) *
FA Mean (SD)	89.7 (18.6)	84.72 (19.99)	86.57 (17.17)	84.65 (15.84)	77.78 (23.79)	78.79 (27.48)	76.48 (17.2) *	84.74 (17.87)	81.30 (18.23) *	79.80 (18.28) *	74.24 (22.03)	81.02 (14.45)	79.58 (22.98)	83.68 (18.89) *	83.35 (19.14) *	83.33 (22.36)	83.53 (17.69) *
REB Mean (SD)	92.5 (18.6)	96.74 (7.87)	91.3 (20.61)	97.61 (6.07)	88.69 (16.68)	100 (0)	79.63 (23.85) *	89.78 (19.19)	85.33 (22.25)	84.09 (22.45)	79.61 (27.33)	100 (0) *	80.95 (27.9)	88.22 (21.07)	90.33 (19.14)	94.79 (14.24)	87.21 (21.13)
PT Mean (SD)	87.8 (19.9)	83.13 (22.65)	83.18 (26.45)	82.86 (22.8)	73.75 (24.94)	69.33 (27.37)	69.38 (26.35) *	79.44 (23.17)	76.05 (25.38) *	73.95 (25.35) *	66.42 (32.99) *	76.26 (21.96)	72.58 (31.22) *	76.29 (26.53) *	77.77 (25.88) *	77.68 (25.65) *	76.1 (26.05) *
PE Mean (SD)	80.3 (19.1)	75.51 (19.82)	73.42 (27.42)	75.88 (22.72)	67.36 (24.48)	66.67 (25.28)	61.63 (25.19) *	67.82 (23.61) *	65.65 (24.16) *	65.57 (23.57) *	47.73 (33.14) *	64.81 (23.12)	54.17 (33.61) *	67.61 (24.72) *	69.2 (24.75) *	68.85 (25.15) *	67.35 (25.02) *
SE Mean (SD)	79.8 (17.5)	73.04 (16.31)	74.78 (17.46)	77.63 (17.14)	74.31 (17.21)	79.92 (15.57)	68.58 (17.45) *	73.08 (18.85)	72.31 (18.13) *	70.36 (16.94) *	65.15 (16.7)	75.93 (18.61)	66.67 (19.78) *	70.83 (17.1) *	72.1 (17.42) *	73.85 (16.82)	70.86 (17.33) *
MH Mean (SD)	78.5 (13.2)	81.36 (16.14)	79.22 (14.68)	78.68 (11.65)	75.42 (16.71)	76.82 (17.5)	74.27 (13.37)	76.00 (13.38)	77.13 (14.66)	75.47 (15.45)	70.45 10.11)	76.11 (6.97)	71.75 (11.84	77.57 (14.69)	78.07 (14.91)	79.44 (15.81)	77.4 (13.96)
BE Mean (SD)	75.6 (16.7)	70.61 (16.58)	74.57 (15.74)	73.42 (13.75)	67.08 (18.02)	67.27 (19.67)	69.38 (13.98)	67.77 (16.3) *	70.24 (15.75)	70.19 (15.9)	66.82 (16.17)	69.44 (22.14)	68.25 (15.58)	69.28 (15.94) *	70.51 (16.16) *	69.69 (16.96)	70.14 (15.76) *
FC Mean (SD)	72.3 (21.6)	68.33 (22.94)	68.62 (22.1)	69.74 (20.98)	67.5 (26.5)	78.64 (15.83)	65.94 (22.64)	66.46 (23.21)	70.12 (21.76)	67.83 (21.54)	63.18 (23.9)	70.56 (19.44)	67.75 (25.26)	69.91 (22.57)	69.71 (22.63)	67.56 (23.15)	71.1 (22.23)
PhS Mean (SD)	53 (8.8)	54.14 (4.3)	52.14 (8.49)	53.82 (3.32)	48.4 (7.31)	50.36 (8.68)	45.03 (11.25) *	50.84 (8.36)	48.06 (10.45) *	47.00 (11.06) *	47.99 (10.38)	50.56 (4.41)	47.90 (11.87)	50.38 (9.24)	50.98 (8.57)	51.87 (8.63)	49.79 (8.99) *
PsS Mean (SD)	51.2 (9.1)	49.52 (8.96)	49.61 (10.01)	50.2 (7.58)	46.84 (7.86)	48.07 (9.73)	45.04 (8.36) *	47.04 (9.09) *	47.10 (8.99) *	46.46 (8.73) *	41.20 (11.68) *	48.19 (8.48)	43.22 (12.32) *	46.87 (9.36) *	47.68 (9.39) *	48.26 (9.32)	46.94 (9.08) *

Abbreviations: CDI-TW total score (CDI-TW), negative mood (NM), interpersonal problems (IP), ineffectiveness (IN), anhedonia (AN), negative self-esteem (NE), pure atrial septal defect (pASD), pure ventricular septal defect (pVSD), pure right-side great arteries anomalies (pRGA), pure left-side great arteries anomalies (pLGA), pure patent ductus arteriosus (pPDA), multiple/complex CHDs with atrial septal defect (mASD), multiple/complex CHDs with ventricular septal defect (mVSD), multiple/complex CHDs with right-side great arteries anomalies (mRGA), multiple/complex CHDs with left-side great arteries anomalies (mLGA), multiple/complex CHDs with venous return anomalies (mVR), multiple/complex CHDs with patent ductus arteriosus (mPDA) cyanotic heart diseases (CyHD), patient had admitted to ICU (ICU), patients had received cardiac catheterization (CC), patients had received therapeutic catheterization (TCC), patients had received operations (OP), physical functioning (PF), role/social-physical (RP), general health perceptions (GH), bodily pain (BP), role/social emotional/behavioral (REB), parental impact-time (PT), parental impact-emotional (PE), self-esteem (SE), mental health (MH), behavior (BE), family activities (FA), family cohesion (FC), change in health (CH), physical summary score (PhS), and psychosocial summary score (PsS). * *p* ≤ 0.001.

**Table 4 children-09-00578-t004:** Multivariant analysis for the relationships between multiple factors and Children Depression Inventory-TW total score, physical summary score, or psychosocial summary score in Child Health Questionnaire—Parent Form 50.

	CDI-TW			PhS			PsS		
	Beta	(SE)	*p*-Value	Beta	(SE)	*p*-Value	Beta	(SE)	*p*-Value
Age	−0.099	0.162	0.539	0.111	0.144	0.441	−0.108	0.172	0.531
Male gender	−0.884	1.061	0.405	0.033	0.942	0.972	−0.765	1.128	0.498
ASD vs Non-ASD	1.535	2.260	0.498	−0.385	2.006	0.848	−1.481	2.402	0.538
pASD vs other CHDs	0.741	2.464	0.764	−4.756	2.187	0.031 *	0.652	2.619	0.804
VSD vs Non-VSD	−4.059	1.934	0.037*	−3.103	1.716	0.072	2.874	2.055	0.163
pVSD vs other CHD	4.130	2.927	0.159	−2.457	2.598	0.345	−3.582	3.111	0.251
RGA vs non-RGA	1.272	2.076	0.541	−3.058	1.843	0.098	−2.342	2.207	0.289
pRGA vs other CHDs	−1.077	3.369	0.749	−1.075	2.990	0.719	0.590	3.580	0.869
LGA vs Non-LGA	3.625	3.258	0.267	−3.877	2.892	0.181	−5.033	3.463	0.147
pLGA vs other CHDs	−2.350	4.592	0.609	−5.263	4.076	0.198	0.918	4.881	0.851
VR vs Non-VR	−4.357	6.266	0.488	2.473	5.561	0.657	7.609	6.660	0.254
PDA vs Non-PDA	−0.203	2.472	0.935	−2.826	2.194	0.199	−3.420	2.627	0.194
pPDA vs other CHDs	4.892	4.513	0.279	−4.475	4.005	0.265	−1.000	4.796	0.835
CyHD vs Non-CyHD	5.616	1.952	0.004 **	−7.411	1.727	0.000 ***	−4.487	2.074	0.031 *
ICU vs Non-ICU	1.423	1.860	0.445	−3.547	1.648	0.034 *	−3.977	1.977	0.045 *
TCC vs non-TCC	−0.537	1.584	0.735	−0.793	1.403	0.587	1.026	1.684	0.543
OP vs Non-OP	1.271	1.922	0.509	1.979	1.699	0.269	0.876	2.042	0.668

Abbreviation: CDI-TW total score (CDI-TW), physical summary score (PsS), psychosocial summary score (PhS), atrial septal defect (ASD), pure atrial septal defect (pASD), ventricular septal defect (VSD), pure ventricular septal defect (pVSD), congenital heart disease (CHD), right-side great arteries anomalies (RGA), pure right-side great arteries anomalies (pRGA), left-side great arteries anomalies (LGA), pure left-side great arteries anomalies (pLGA), venous return anomalies (VR), patent ductus arteriosus (PDA), pure patent ductus arteriosus (pPDA), cyanotic heart diseases (CyHD), the patient had admitted to ICU (ICU), patients had received therapeutic catheterization (TCC), patients had received operations (OP).* *p* ≤ 0.05. ** *p* ≤ 0.01. *** *p* ≤ 0.001.

**Table 5 children-09-00578-t005:** Multivariant analysis for the relationship between physical summary score, and psychosocial summary score in Child Health Questionnaire—Parent Form 50.

	CDI-TW Score		
	Beta	(SE)	*p*-Value
PhS	0.01	0.06	0.852
PsS	−0.33	0.06	<0.0001 *

Abbreviation: CDI-TW total score (CDI-TW), physical summary score (PsS), psychosocial summary score (PhS), * *p* ≤ 0.001.

## Data Availability

The data used in this study are available in Appendix A.

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
