# Peer review of "Factors Affecting Psychological and Health-Related Quality-of-Life Status in Children and Adolescents with Congenital Heart Diseases"

_children, 2022, doi:10.3390/children9040578_

Round 1

Reviewer 1 Report

The manuscript entitled"Factors affect psychological and health-related quality-of-life status in children and adolescents with congenital heart diseases" is a very significant study of CHD (congenital heart disease) and its relation with psychological and HRQoL (health related quality of life). Significant number of patients has been studied for this and results were clearly established with different parameters. It would  be better if the authors can analyzed the data from the samples collected from a different period of time (1year) instead of short 5 month span.

Author Response

Thank you so much for your comment. Our study design is a randomized cross-sectional study to prevent bias. Our sample selection aim was to collect a wide range of different congenital heart diseases. The target patients are school-age congenital heart disease (CHD) children and adolescents who might receive operations or treatments at an early age.

For most school-aged moderate and severe CHD children and adolescents, in order to match their school schedule, they will visit our hospital during their summer vacation, which is the period we covered in our sample collection. The time of 5 months may seem like a short period, but it contains the most variety of CHDs in our hospital, especially severe CHDs. If we increase the period to a longer time, for example, one year, in order to achieve random selection, we will collect too many simple CHDs but a small portion of moderate and especially severe CHDs. Therefore, the number of moderate and severe CHDs will be too small to analyze.

Reviewer 2 Report

The work “Factors affect psychological and health-related quality-of-life status in children and adolescents with congenital heart diseases” explores the relevant question about psychological well-being and quality of life of children with congenital heart disease, especially if you take into account the high incidence of this disease.

The work is interesting and well-conducted, but I have some concerns and advice:

CDI mean scores did reveal clinical elevations only for the group with the most severe condition. However, reporting the number of individual patients with significant symptoms could offer a more realistic picture of the situation. In fact, often mean values mask relevant information: the CDI scores reported in Table 2 showed a very large SD, that are in some cases of the same dimension of the mean, indicating an extreme variability between patients and suggesting the presence of pathological conditions for many individuals. A solution could be to indicate the number and percentage of patients with clinically relevant symptoms for each group. Furthermore, it could also be that the use of population norms is a less sensitive procedure. In a previous study of our group (Cainelli et al., “Detecting neurodevelopmental trajectories in congenital heart

diseases with a machine‑learning approach”), we found a similar result evaluating neuropsychology and psychopathology in CHD children: all mean CHD scores were still within the average range in terms of population norms, but by comparing with a control group, several differences emerged. One reason may be that a healthy control group of typically developing children, growing up in the same period, offered a more representative reflection of normal variation. Another reason may be that the group’s comparisons allowed us to highlight subclinical vulnerabilities.

Finally, the CDI could have too poor sensitivity to subtle and non-clinical conditions, highlighting only the most severe cases. The authors could hypothesize for future research to insert more broad measures of psychological functioning (also simply like CBCL).

Minor:

Exclusion criteria: “mentally handicapped or intellectual disability”: I think the concepts are overlapping…mentally handicapped is not politically correct; I would leave only intellectual disability

The text appears poorly edited; some examples:

Page 2, line 54: incorrect capitalization

Page 2, lines 84-84: incorrect syntactical construction of the sentence

Page 9, lines 239-240: incorrect syntactical construction of the sentence

Author Response

  1. Thank you so much for your comment. In your paper “Detecting neurodevelopmental trajectories in congenital heart diseases with a machine‑learning approach”, there could be variations within CHD groups. In order to give a more explicit reference, we have updated table 2 by adding the number and percentage of depressed patients in each CHD group. We also updated our manuscript and emphasized CDI’s limitations in line 398: "In addition, CDI-TW could only pick up the most severely depressed patients but have poor sensitivity to detect subtle and non-clinical conditions”. We also highlighted the future research suggestions in line 404-406 with “Further investigations could focus on the impact of QoL and depressive conditions on different CyHDs by using a cardiac-specific tool and insert more broad measurements for psychological functioning”.
  2. Thank you so much for your suggestion. We have updated our manuscript and removed “mentally handicapped” in our exclusion criteria in line 89.
  3. Thank you for your careful review. We have re-edited our whole manuscript and corrected multiple mistakes. For example:
    1. Line 54: Jackson et al. described chronic emotional distress might increase CV complications and premature mortality “in” ACHDs
    2. Line 82-84: Three hundred and fifty patients with structural CHDs visited our hospital as an in- or out-patient “from 1st May to 31st September 2018 and one of their major caregivers was asked for participation”
    3. Line 235-236: “Having CyHD and VSD significantly affected patients’ depressive condition (CDI-TW) among all CHD patients in multivariant analysis (Table4)”
